# Untranslated Region Sequences and the Efficacy of mRNA Vaccines against Tuberculosis

**DOI:** 10.3390/ijms25020888

**Published:** 2024-01-10

**Authors:** Vasiliy Reshetnikov, Ilya Terenin, Galina Shepelkova, Vladimir Yeremeev, Semyon Kolmykov, Maxim Nagornykh, Elena Kolosova, Tatiana Sokolova, Olga Zaborova, Ivan Kukushkin, Alisa Kazakova, Dmitry Kunyk, Anna Kirshina, Olga Vasileva, Kristina Seregina, Ildus Pateev, Fedor Kolpakov, Roman Ivanov

**Affiliations:** 1Translational Medicine Research Center, Sirius University of Science and Technology, 354340 Sochi, Russia; 2Institute of Cytology and Genetics, Siberian Branch of Russian Academy of Sciences, 630090 Novosibirsk, Russia; 3Belozersky Institute of Physico-Chemical Biology, Lomonosov Moscow State University, 119991 Moscow, Russia; 4Central Tuberculosis Research Institute, 107564 Moscow, Russia

**Keywords:** 5′UTR, 3′UTR, mRNA therapeutics, translation efficiency

## Abstract

mRNA vaccines have been shown to be effective in combating the COVID-19 pandemic. The amount of research on the use of mRNAs as preventive and therapeutic modalities has undergone explosive growth in the last few years. Nonetheless, the issue of the stability of mRNA molecules and their translation efficiency remains incompletely resolved. These characteristics of mRNA directly affect the expression level of a desired protein. Regulatory elements of RNA—5′ and 3′ untranslated regions (UTRs)—are responsible for translation efficiency. An optimal combination of the regulatory sequences allows mRNA to significantly increase the target protein’s expression. We assessed the translation efficiency of mRNA encoding of firefly luciferase with various 5′ and 3′UTRs in vitro on cell lines DC2.4 and THP1. We found that mRNAs containing 5′UTR sequences from eukaryotic genes *HBB*, *HSPA1A*, *Rabb*, or *H4C2*, or from the adenoviral leader sequence TPL, resulted in higher levels of luciferase bioluminescence 4 h after transfection of DC2.4 cells as compared with 5′UTR sequences used in vaccines mRNA-1273 and BNT162b2 from Moderna and BioNTech. mRNA containing TPL as the 5′UTR also showed higher efficiency (as compared with the 5′UTR from Moderna) at generating a T-cell response in mice immunized with mRNA vaccines encoding a multiepitope antigen. By contrast, no effects of various 5′UTRs and 3′UTRs were detectable in THP1 cells, suggesting that the observed effects are cell type specific. Further analyses enabled us to identify potential cell type-specific RNA-binding proteins that differ in landing sites within mRNAs with various 5′UTRs and 3′UTRs. Taken together, our data indicate high translation efficiency of TPL as a 5′UTR, according to experiments on DC2.4 cells and C57BL/6 mice.

## 1. Introduction

Therapeutics based on mRNA have proven to be a promising option in vaccination, gene therapy, and cancer immunotherapy. The advantages of mRNA therapeutics include their high efficiency, reduced frequency and severity of adverse reactions, and relatively low production costs. These advantages explain the large number of preclinical and clinical trials of mRNA therapeutics in the past five years. Nevertheless, mRNA therapeutics require the optimization of mRNA sequences because their effectiveness primarily depends on the rate of degradation (and the rate of translation initiation) of mRNA [1,2]. 

Untranslated regions (UTRs) are noncoding portions of the sequence of mRNA located upstream (5′UTR) and downstream (3′UTR) of the coding region of mRNA. It is known that UTRs are associated with processes of mRNA translation, and by interacting with RNA-binding proteins (RBPs) and microRNAs, UTRs can influence mRNA’s translation efficiency and resistance to degradation [3,4,5,6,7]. In the 5′UTR portions, among the specific sequences that affect translation efficiency, there are the terminal oligopyrimidine tract (TOP) sequence, consisting of 7–14 residues of cytosine and thymine forming a TCT motif; internal ribosome entry sites (IRESs); cap-independent translation enhancers (CITEs); upstream open reading frames (uORFs); and protein-binding motifs [2]. There are at least two pathways (cap-dependent and cap-independent) that trigger translation through these elements, and each pathway requires certain *trans*-acting proteins; the efficiency of the triggering of translation via these mechanisms depends on the cellular state [2]. 3′UTR AU-rich elements can determine the rate of mRNA decay under various conditions [1,8]. 

Globin gene sequences are often used to ensure high translation efficiency. In particular, sequences of human α- and β-globin [9,10] and rabbit β-globin [10,11] are employed to improve the translation efficiency of heterologous mRNAs. The 5′UTR of rabbit β-globin mRNA is thought to mediate efficient cap-dependent translation and does not contain any secondary structures or sequence elements that may negatively affect translation [10]. Aside from the regulatory sequences of human globin genes, 5′UTR sequences of heat shock protein HSP70 and of histone H3 are also utilized as heterologous UTRs [12,13]. The effect of *HSP70* 5′UTR has been observed in several cell lines [14], indicating that this effect is not cell type specific. Some regulatory elements, such as adenovirus TPL (tripartite leader), can enhance protein expression by enhancing translation initiation [15]. To sum up the data on natural 5′UTRs, one can conclude that regulatory sequences of constitutively expressed genes are often used to enhance the translation efficiency of heterologous mRNAs.

There are much fewer studies assessing the contribution of heterologous 3′UTRs to the level of a desired protein’s expression as compared with studies assessing the effects of 5′UTRs. Regulatory sequences of human globin genes are also used as 3′UTRs to ensure the high stability and translational activity of mRNA [16]. A systematic study of the influence of 3′UTRs on stability and translation efficiency, using bioinformatic techniques, was carried out by Orlandini von Niessen et al. [17]. They showed that 3′UTRs containing mtRNR1 and AES promote the expression of the upstream gene. MtRNR1 is mitochondrial noncoding 12S ribosomal RNA (rRNA) [18]. AES is a distinct member of the family of Groucho/Transducin-like enhancers of split genes; it regulates androgen receptor genes’ transcriptional activity and Notch and Wnt signaling, and acts as a tumor suppressor [19].

In addition to the use of natural regulatory sequences, active research is currently being carried out for optimal, artificial (not occurring naturally) sequences of UTRs [20].

Finding optimal combinations of 5′UTRs and 3′UTRs that are more effective may improve the efficacy of mRNA therapeutics, help to lower their dose, and reduce adverse effects. Despite the numerous preclinical and clinical trials of mRNA therapeutics, the number of studies on the impact of regulatory sequences on the translation and stability of heterologous RNAs is limited at present. Therefore, in our work, we assessed the influence of different types of 5′ and 3′UTRs on luciferase mRNA expression in cell lines DC2.4, and THP1. Furthermore, we evaluated the impact of TPL as the 5′UTR on the efficacy of a multiepitope mRNA vaccine against *Mycobacterium tuberculosis*.

## 2. Results

### 2.1. The Effect of the 5′UTR Sequence on Translation Efficiency

We compared mRNA translation efficiency levels among 10 luciferase-coding transcripts in two cell lines (THP1 and DC2.4; Figure 1). The Kruskal–Wallis test showed an effect of the 5′UTR sequence on luciferase bioluminescence intensity in mouse dendritic cell line DC2.4 at 4 h after transfection [Kruskal–Wallis test: H (11, N = 66) = 85.2, *p* < 0.0001] and in human monocyte cell line THP1 at 24 h after transfection [Kruskal–Wallis test H (11, N = 60) = 31.05, *p* = 0.001; Figure 2]. 5′UTRs from mRNAs of eukaryotic genes *HBB*, *HSPA1A*, *Rabb*, and *H4C2* and the adenoviral 5′UTR termed “tripartite leader sequence” (TPL) significantly increased protein levels 4 h post-transfection in DC2.4 cells as compared with the 5′UTRs used in the BNT162b2 (Pfizer-BioNTech) and mRNA-1273 (Moderna) vaccines and compared with the 5′UTR from complement gene *C3* (Figure 2). 

By contrast, at later time points after the transfection in DC2.4 cells these differences disappeared. In the THP1 cell line, mRNA with the 5′UTR sequence from human β-globin mRNA also showed elevated luciferase levels 24 h after the transfection as compared with mRNA with the 5′UTR sequence from *C3* mRNA. 

The observed differences among the cell lines in translation efficiency of Luc RNAs containing various 5′UTRs suggested that the effects of 5′UTRs were distinctly cell type specific.

### 2.2. The Impact of the 3′UTR Sequence on Translation Efficiency

Similarly to the results on the 5′UTR, differences in bioluminescence intensity after transfection of Luc RNAs containing different 3′UTRs were observed only in the DC2.4 cell line (Figure 3). In this context, an effect of the 3′UTR on luciferase levels was registered 8, 24, and 72 h after the transfection of DC2.4 cells [Kruskal–Wallis test 8 h: H (11, N = 66) = 38.37, *p* < 0.001; 24 h: H (11, N = 66) = 38.49, *p* < 0.001; and 72 h: H (11, N = 66) = 30.49, *p* = 0.001]. Sequences AES-HBB and mtRNR-HBB, when used as the 3′UTR in Luc RNA, yielded the lowest level of luciferase (i) as compared with 3′UTR sequences AES-AES and mtRNR-mtRNR at 8 h after the transfection (*p* < 0.05), (ii) as compared with 3′UTR sequence AES-EMCV at 8 and 24 h after the transfection (*p* < 0.05), and (iii) as compared with the 3′UTR from mRNA-1273 at 24 an 72 h after the transfection. Because the 3′UTR is primarily responsible for RNA stability in the cell and prevents the RNA degradation mediated by binding to microRNA, our findings implied that 3′UTR sequences AES-HBB and mtRNR-HBB within mRNA lead to a decrease in its stability. On the other hand, none of the tested 3′UTR sequences raised luciferase levels as compared with the control Luc RNAs containing 3′UTR sequences from mRNA-1273 and BNT162b2. Even the 3′UTR sequence AES-EMCV—which caused higher relative bioluminescence as compared with the mRNA-1273 3′UTR in Luc RNA at 8 and 24 h after the transfection of DC2.4 cells—showed lower bioluminescence values 72 h after the transfection of DC2.4 cells as compared with the mRNA-1273 3′UTR (*p* < 0.05). 

There were no significant differences in the THP1 cell line. However, in DC2.4 data sequences, AES-AES, AES-EMCV, and AES-HBB, when used as the 3’ UTR in Luc RNA, resulted in the lowest levels of luciferase compared with the BNT162B2 and mRNA-1273 data sequences.

### 2.3. Cell Type-Specific Features of Translational Machinery of THP1 and DC2.4 Cells

Next, we tried to answer the question of which features of the translational machinery of cell lines can explain the difference in the observed effects of 5′UTR and 3′UTR composition on translation efficiency. For this purpose, by means of publicly available transcriptomic data, we assessed the expression levels of *RBP* genes in cell lines THP1 and DC2.4. We revealed that the expression of 85 *RBP* genes differs between THP1 and DC2.4 cells (Appendix A). Twenty differentially expressed *RBP* genes were found to have a presumed landing site in the UTRs under study (Table 1). 

### 2.4. Unique Features of the 5′UTR and 3′UTR Sequences

We attempted to comprehensively characterize our selected sequences. For instance, for 5′UTR sequences, mean ribosomal loading (MRL) was calculated by bioinformatic algorithms (Table 1). For 3′UTR sequences, Gibbs free energy was computed (Table 1). The values of MRL and Gibbs free energy were then analyzed for correlation with data on relative levels of bioluminescence after the transfection of the cell lines. In addition, in all our 5′UTR and 3′UTR sequences, a search for the binding sites of RBPs was carried out.

Our results uncovered a significant correlation of MRL with bioluminescence intensity at 4 h after transfection in DC2.4 cells (r^2^ = 0.43, *p* = 0.024; Figure 4) and a marginally significant correlation at later time points (24 and 72 h after transfection: r^2^ = 0.3 and r^2^ = 0.29). In the THP1 cell line, however, there was no correlation between these parameters. We also failed to find a correlation between minimum free energy (MFE) (of either a 3′UTR alone or the entire RNA) and bioluminescence intensity (Appendix A).

Next, we searched for binding sites for known RBPs (*p* > 0.0001, Table 1). Most of the 5′UTR and 3′UTR sequences were found to contain the predicted binding sites for various RBPs, many of which directly affect the stability and translation efficiency of mRNA (Table 2). Nonetheless, a direct relation between the presence of a landing site and the RNA stability was difficult to discern because some RBPs stabilize RNA, some destabilize it, and other RBPs can regulate RNA stability in a context-dependent manner, as is the case for PCBP2 (Table 2). It should also be pointed out that the effect of most RBPs on the stability and translation efficiency of mRNA has been poorly investigated and apparently may strongly depend on both the binding region (5′UTR, ORF, or 3′UTR) and the presence of other auxiliary proteins. Of note, three of the top five UTRs (from *HBB*, *Rabb*, and *H4C2* mRNAs) do not have RBP-binding sites, and the other two (TPL and HSPA1A) carry binding sites for the RBPs whose influence on RNA stability is not clear. Regarding 3′UTRs, only the sequence mtRNR-mtRNR and the mRNA-1273 3′UTR, both of which produced the best results, contain a predicted binding site for an RBP, HNRNPK, which is associated with the stabilization of mRNA. 

### 2.5. In Vivo Validation of Efficiency of the TPL Sequence as a 5′UTR for Vaccine Development

At the next step, we chose TPL as the 5′UTR sequence because it showed the best results in the DC2.4 cell line, and had best MRL, and compared it with the 5′UTR from mRNA-1273 in an in vivo experiment on C57BL/6Cit and I/StSnEgYCit mice, which are extremely susceptible to tuberculosis [33]. Sequences of the 5′UTRs were tested in constructs encoding a multiepitope mRNA vaccine against M. tuberculosis described previously [33]. 3′UTR from mRNA-1273 was used in these constructs. 

The ELISpot assay proved the vaccination to be effective [Kruskal–Wallis test: H(3, 18) = 14.82, *p* = 0.002]. The number of IFNγ-secreting cells after stimulation with the *M. tuberculosis* sonicate was higher for splenocytes of mice vaccinated with the mRNA vaccine containing the TPL sequence as the 5′UTR compared with splenocytes of mice from control groups (PBS or LNP wo RNA, *p* < 0.01; Figure 5D). As for the mRNA vaccine containing the mRNA-1273 5′UTR, the respective experimental group had a numerically larger number of IFNγ-secreting cells compared with the control group (5.6 ± 0.7 vs. 1.3 ± 0.6) or the LNP group (5.6 ± 0.7 vs. 1.0 ± 0.4), but these differences were not significant. A comparison of two mRNA vaccines showed that mRNA with the 5′UTR TPL sequence gives a more than two-fold increase in the number of IFNγ-secreting cells after the immunization as compared with the mRNA with the mRNA-1273 5′UTR sequence (13.6 ± 1.9 versus 5.6 ± 0.7). 

The vaccination led to the development of the delayed-type hypersensitivity reaction [Kruskal–Wallis test: H(3,31) = 9.07, *p* = 0.028]. Mice immunized with the mRNA vaccine containing the TPL sequence as the 5′UTR showed marked enlargement of the site of tuberculin injection as compared with the group of mice injected with LNPs (*p* < 0.05). On the other hand, mice immunized with the mRNA vaccine containing the mRNA-1273 sequence as the 5′UTR did not differ from control groups.

Finally, vaccination with two doses of mRNA vaccines had an effect on the of *M. tuberculosis* load in lungs 50 days after the infection [Kruskal–Wallis test: H(3,20) = 12.58, *p* = 0.006]. In lung tissue, there was a lower number of colony-forming units (CFUs) in mice immunized with mRNA vaccine 5′-TPL-mEpitope-mRNA1273-3′ in comparison to the PBS and LNP groups (*p* < 0.05). Mice immunized with mRNA vaccine 5′-mRNA1273-mEpitope-mRNA1273-3′ also exhibited a >1.5-fold lower bacterial load as compared with the control groups, but the differences were statistically insignificant (*p* > 0.05).

Taken together, our results indicate that the 5′-TPL-mEpitope-mRNA1273-3′ vaccine induced stronger adaptive immunity and protection against *M. tuberculosis*. Thus, the TPL sequence as a 5′UTR can significantly improve the effectiveness of mRNA therapeutics.

## 3. Discussion

In this work, we found that the use of firefly luciferase mRNA carrying a 5′UTR sequence from the mRNA of *HBB*, *HSPA1A*, *Rabb*, *H4C2*, or *TPL* results in higher bioluminescence intensity after transfection of the DC2.4 dendritic cell line in comparison with known sequences of the 5′UTR from mRNA-1273 and BNT162b2 mRNA vaccines. We performed in silico analysis of these sequences and predicted MRL values, which correlated with the bioluminescence data in the cell-based assay. Next, as a 5′UTR, we chose the TPL sequence, which had high efficiency in cells and the highest MRL values, and tested it as part of a multiepitope mRNA vaccine against *M. tuberculosis*. 

There are two experimental studies that demonstrated efficacy of the mRNA vaccines against tuberculosis [34,35]. Immunization with an mRNA vaccine based on the full-length MPT83 antigen triggered production of specific antituberculosis antibodies [35]. Another mRNA vaccine contains four full-length tuberculosis antigens fused with a synthetic TLR4 agonist [34]. Our study is the first one to compare mRNA vaccines with various 5′UTRs; we have shown that use of TPL as a 5′UTR compared with the 5′UTR from mRNA-1273 evoked more pronounced protective adaptive immune responses in mice leading to higher preclinical efficacy of mRNA vaccines against tuberculosis. Further optimization of the mRNA tuberculosis vaccine may include increasing the number of epitopes encoded, changing the type of cap (from cap0 to cap1) and adjusting the dose. Use of HBB as a 5′ UTR also deserves further investigation.

TPL is a nucleotide sequence in the 5′UTR of adenovirus late mRNAs and has previously been thought to initiate translation of late adenoviral mRNAs in a cap-independent manner [15,36]. TPL contains an internal ribosome entry site enabling it to recruit a ribosome independently of interactions with the mRNA 5′ 7-methyl G cap. Nonetheless, the contribution of this alternative mechanism of translation initiation to our results appears to be negligible because the theoretical amount of capped RNA was ~80%, and cap-independent translation plays a major role only during periods of cellular stress, apoptosis, and certain stages of the cell cycle [2]. Furthermore, although it is currently unclear whether the TPL sequence is capable of initiating translation in a cap-independent manner, evidence from our research group suggests that TPL does not function as an internal ribosome entry site (unpublished data).

Similarly to TPL, 5′UTRs from mRNAs of *HSPA1A*, *Rabb*, *HBB*, and *H4C2—* when tested as part of mRNA—yielded higher translation efficiency in cell experiments as compared with the mRNA-1273 5′UTR. On the other hand, TPL as the 5′UTR had the highest MRL (7.7); this parameter evaluates experimental data on ribosomal loading by means of neural networks [37]. On the basis of 280,000 different 5′UTR sequences, the Sample et al. [37] study demonstrated that MRL values range from two to 11; however, among the highest MRLs, only in the range six–eight was there a high correlation (r^2^ = 0.77) between the predicted and observed values. Of note, other 5′UTR sequences (from *Rabb*, *HSPA1A*, *HBB*, and *H4C2*) that manifested higher translation efficiency (when tested as part of mRNA), as compared with the mRNA-1273 5′UTR, also had higher MRL than the mRNA-1273 5′UTR sequence did (6.59, 6.58, 6.52, and 6.17 versus 5.74). Therefore, these findings indicate that the TPL 5′UTR sequence contains apparently unique sequences that are associated with high ribosome loading.

We hypothesized that such effects could be explained by the presence of binding sites for RBPs. Such proteins, while interacting with certain motifs of RNA molecules, can both directly influence the future fate of RNA and can do so indirectly by attracting various protein complexes [14]. In the 5′UTR sequences of *HBB*, *HSPA1A*, *Rabb*, and *H4C2*, and in TPL, binding sites for only two RBPs (PABPC3 and RBM22) were found; however, the function of these RBPs is related to either ensuring the stability of the poly(A) tail or regulation of alternative splicing and plays a negligible role in the translation of heterologous RNAs. On the other hand, the 5′UTR from mRNA-1273 has a predicted landing site for HNRNPH1; the binding of this RBP is associated with higher RNA stability [30]. Nevertheless, it should be noted that the direction of the influence of an RBP on the stability and translation efficiency of RNA can depend on the cell type and on the target RNA, as exemplified by PCBP2: it can stabilize some mRNAs but promotes the degradation of other mRNAs [25,26,27]. In addition, RBPs can compete for binding to microRNA, thereby having either synergistic or opposing effects on RNA stability [38,39,40].

Despite pronounced differences in translation efficiency among the tested 5′UTRs in the DC2.4 cell line, these differences were absent in the THP1 cell line, indicating that this effect is cell type specific. To answer the question which cell type-specific features of the translational machinery of THP1 and DC2.4 cells may be responsible for the observed effect, we identified 85 genes of RBPs differentially expressed between the cell lines, thus highlighting differences in their translational machinery. Only for eight differentially expressed genes of RBPs is the influence of such proteins on RNA stability known (Table 2). Nonetheless, effects of many RBPs on translation are yet to be identified [41]. RBPs monitor mRNA stability and decay and control the correct translation of mRNA. Furthermore, the dynamic association of RBPs with mRNA monitors the localization of mRNA, in particular its movements between actively translating ribosomes, stress granules, and *p* bodies in a context-dependent manner [42]. The transport function of RBPs can be mediated either by their direct binding to motor proteins or, indirectly, through motor protein cargo adaptors [42]. 

Our cell type-specific effects highlight the importance of evaluating various regulatory sequences in different models, both cellular and animal ones. The assessment of translational machinery may be a promising tool for the development of preventive and therapeutic mRNA modalities, for example, via activation/suppression of translation in specific cell types. Nevertheless, it is obvious that the assessment of translational machinery should not be limited to the evaluation of RBPs’ expression; it is also necessary to assess the expression of regulatory RNAs (among which microRNAs play the most prominent part) as well as a ratio between these components.

Unexpectedly, there were no significant differences between the 3′UTR sequences tested in our cell-based assay, and none of the 3′UTR sequences outperformed the controls. The lack of pronounced differences between different 3′UTR sequences may be due both to peculiarities of the translational machinery and to assay limitations. In order to assess the efficacy of 3′UTRs, Orlandini von Niessen et al. [16] also used reverse transcription followed by PCR to quantify the RNA itself, as well as a technique for preparing induced pluripotent stem cell colonies from human neonatal fibroblasts. The relatively short half-life of luciferase may have an impact on the results [43], preventing the detection of differences in the cumulative transgene expression. It should be noted that a definitive conclusion regarding the efficiency of UTRs should be made based on direct or indirect evaluation of target antigen expression, since translational requirements may vary for different transgenes. Further research in animal models is needed to find the reasons for the functional differences between the 3′UTRs we selected. 

Although we have demonstrated the stability of the 5′-TPL-mEpitope-mRNA1273-3′ vaccine for at least 6 months when stored at +2 to +8 °C, the impact of UTR selection on the stability of mRNA should be evaluated further. Engineering of mRNA vaccines with more favorable storage conditions and longer shelf life may increase their availability in regions with limited cold chain infrastructure and high tuberculosis prevalence. 

## 4. Materials and Methods

### 4.1. Design of In Vitro Experiments

At first, 22 mRNA constructs encoding firefly luciferase were assembled with various combinations of 5′ and 3′UTRs. 5′UTRs from mRNAs of genes *CFL1*, *C3*, *HSPA1A*, *TPL*, *GAPDH*, *FH1*, *HBB*, *Rabb*, *H4C2*, and *eEF2* were chosen as 5′UTRs in combination with the 3′UTR from mRNA-1273 (Moderna, Cambridge, MA, USA). The 5′UTR from mRNA-1273 was combined with various 3′UTRs, such as AES-AES, AES-HBB, AES-EMCV, mtRNR, mtRNR-mtRNR, mtRNR-AES, mtRNR-HBB, mtRNR-EMCV, HBB, and HBB-EMCV (Figure 1). Two other constructs were prepared that contained 5′ and 3′UTRs from either mRNA-1273 or BNT162b2 (Pfizer BioNTech, New York, NY, USA). Sequences of the regulatory regions of these vaccines served as a reference. However, it should be noted that, unlike in the original studies, in our work unmodified uridine was used for mRNA synthesis, not its analogs. Next, complexes of mRNA and a transfection agent were transfected into cell lines DC2.4, and THP1. Finally, luciferase activity was assayed at 4, 8, 24, and 72 h after transfection.

### 4.2. Design of Animal Experiments

C57BL/6JCit and I/StSnEgYCit mice were bred and maintained under conventional, non-SPF conditions with water and food provided ad libitum at the Animal Facilities of the Central Tuberculosis Research Institute (CTRI, Moscow, Russia). 

Female mice of both strains were immunized intramuscularly with one of two versions of mRNA vaccines: mRNA 5′-TPL-mEpitope-mRNA1273-3′ or 5′-mRNA1273-mEpitope-mRNA1273-3′, 50 μg of each RNA. Additionally, two control groups were set up: the first one with the injection of lipid nanoparticles (LNPs) without mRNA (LNP wo RNA), and the second one with the injection of a 1× phosphate buffer (pH = 7.5). Three weeks after the first immunization, the animals were immunized again with the same dose of the vaccines (Figure 5). 

C57BL/6JCit: 3 weeks after the second immunization, 4–5 mice from each group were euthanized, and spleens were excised for subsequent ELISpot analysis. The remaining animals from each group (8 per group) were tested for delayed-type hypersensitivity reactions. Namely, 40 μL of PBS containing 50 IU of tuberculin (from the Scientific Centre for Expert Evaluation of Medicinal Products, affiliated with the Ministry of Health of the Russian Federation) was injected into the left hind paws of the mice. At 48 h after the injection, the swelling of the paws was measured. Data are presented as Δ (swelling of the left paw minus swelling of the right paw).

Tuberculosis-susceptible I/StSnEgYCit mice were infected with virulent strain H37Rv of *M. tuberculosis* at a dose of 500,000 colony-forming units (CFUs) per mouse 3 weeks after the second vaccination. Fifty days after the infection initiation, mycobacterial load of the spleen and lungs was assessed (in 5 mice per group).

### 4.3. Cloning

Cloning of constructs for subsequent in vitro transcription of RNA was carried out using a commercial vector (pSmart; Lucigen, Middleton, WI, USA). The sequence of firefly luciferase (1656 bp) served as a reporter gene. Various eukaryotic and viral sequences (Appendix A) were used as 5′ and 3′UTRs, which were assembled by PCR involving overlapping oligonucleotides. Insert 5′UTR-FFLuc-3′UTR was cloned into the vector via restriction sites *Eco*RI and *Bgl*II. Downstream of the 3′UTR, there was a poly(A) tail sequence consisting of 110 adenines. The resulting vectors were used to transform NEB-stable cells (New England Biolabs, Hitchin, UK), which were cultured at 30 °C and 180 rpm. Colonies were checked for the presence of the insert by PCR, and sequence correctness was confirmed by Sanger sequencing. Isolation and purification of plasmid DNA were performed by means of the Plasmid Miniprep Kit (Evrogen, Moscow, Russia). To linearize the plasmid, the plasmid sample was digested with the *Spe*I restriction enzyme at a unique restriction site located immediately after the poly(A) tail.

The vaccine constructs pSmart-TPL-mEpitope-mRNA1273 and pSmart-mRNA1273-mEpitope-mRNA1273 for the in vivo experiment were also assembled based on the pSmart vector (Lucigen, Middleton, WI, USA). Sequences of the regulatory regions (5′ and 3′UTRs) were fused with the coding sequence by the overlap extension PCR method. The coding sequence has been described previously [44] and consists of 408 bp, including two stop codons and five epitopes of a secretory protein of *M. tuberculosis:* ESAT6 (rv3875). The cloning was also performed at the *Eco*RI and *Bgl*II sites. After the 3′UTR, there was a sequence of a poly(A) tail composed of 110 adenines and a unique restriction site, *Spe*I, at which the vector was linearized for in vitro transcription.

### 4.4. In Vitro Transcription

This procedure was performed as described previously [45]. The reaction mixture contained a buffer consisting of 20 mM DTT, 2 mM spermidine, 80 mM HEPES-KOH pH 7.4, 24 mM MgCl_2_, 3 mM of each ribonucleoside triphosphate (Biosan, Novosibirsk, Russia) and 12 mM of a cap analog called ARCA (Biolabmix, Novosibirsk, Russia). The remaining components—per 100 μL of the reaction mixture—were as follows: 40 U of the RiboCare ribonuclease inhibitor (Evrogen, Moscow, Russia), 500 U of T7 RNA polymerase (Biolabmix, Novosibirsk, Russia), 5 μg of a linearized plasmid, and 1 μL of a mixture of enzymes from a kit called the RiboMAX Large Scale RNA Production System (Promega, Fitchburg, WI, USA) as a source of inorganic pyrophosphatase. The reaction was carried out for 2 h at 37 °C, after which another 3 mM of each ribonucleoside triphosphate was added to the reaction and incubated for an additional 2 h. DNA was hydrolyzed with the RQ1 nuclease (Promega, Fitchburg, WI, USA), and RNA was precipitated by the addition of LiCl to a concentration of 0.32 M and EDTA pH 8.0 to a concentration of 20 mM, followed by incubation on ice for an hour. Next, the solution was centrifuged for 15 min (25,000× *g*, 4 °C). The RNA pellet was washed with 70% ethanol, dissolved in ultrapure water, and precipitated again with alcohol according to the standard procedure. The concentration of RNA was determined spectrophotometrically by means of absorbance at 260 nm. The desired length and homogeneity of the synthesized RNA molecules were verified by capillary electrophoresis on a TapeStation instrument (Agilent Technologies, Santa Clara, CA, USA).

### 4.5. Cell Cultivation and Transfection

DC2.4 and THP1 cells were grown in the RPMI-1640 medium (Capricorn, Ebsdorfergrund, Germany) containing 10% of fetal bovine serum (HyClone, Logan, UT, USA), 50 µM β-mercaptoethanol (Gibco, Grand Island, NY, USA), alanyl-glutamine, and vitamins (PanEco, Moscow, Russia). One day before transfection, DC2.4 cells were seeded in a 48-well plate (0.2 mL of medium per well) at a confluence of 30%. THP1 cells were grown in a suspension culture flask at a concentration of 200,000–800,000 cells per milliliter of the medium. One day before transfection, cells were centrifuged for 5 min and resuspended in a fresh medium; 12-O-tetradecanoylphorbol-13-acetate was added to a concentration of 100 nM to induce differentiation, and the cells were seeded in a standard 48-well plate at a density of 70,000 cells per well. After 12 h, the medium was changed to the one without the phorbol ester.

RNA was diluted to 100 ng/μL. In a separate test tube, a solution of reference mRNA (encoding NanoLuc luciferase) was prepared in sterile 1× phosphate buffer (pH 7.5) to ensure 25 μL of the buffer and 50 ng of NanoLuc mRNA per well. After that, this solution was dispensed into sterile microfuge tubes (25 μL per tube), and the tested mRNAs were introduced (100 ng per tube in the case of DC2.4 cells and 200 ng per tube in the case of THP1 cells). In a separate test tube, a solution of transfection reagent GenJect-40 (Molecta, Moscow, Russia) was prepared in sterile 1× phosphate buffer pH 7.5 to ensure 0.25 μL of the reagent per 100 ng of mRNA being transfected. In accordance with the manufacturer’s protocol, the reagent was incubated in PBS at room temperature for 10 min, after which 25 μL was added to each mRNA solution, mixed, and incubated for another 15 min at room temperature. Next, the transfection mixture was added dropwise to cells, and the cells were incubated for 4, 8, 24, or 72 h.

### 4.6. Dual Luciferase Assay

At 4, 8, 24, and 72 h after transfection, the medium was collected, the cells were washed with 1× phosphate buffer pH 7.5, and the expression of luciferases was analyzed using the Dual Luciferase Assay Kit (Promega, Fitchburg, WI, USA). Luminescence was immediately measured on a CLARIOstar Plus device at wavelengths of 579–679 nm (gain = 3600) for luciferase. The transfection procedures and bioluminescence assessment for each construct were performed in 4–6 biological replicates (with 3 technical repetitions for each). Bioluminescence data from firefly luciferase were normalized to the bioluminescence of NanoLuc luciferase, which served as an internal control.

### 4.7. Predictive Assessment of Mean Ribosomal Loading (MRL) for the Selected 5′UTRs

The Optimus 5-Prime algorithm was employed to predict the impact of the selected 5′UTRs on translation efficiency [37]. Optimus 5-Prime is a method for predicting the MRL of a transcript through the use of convolutional neural networks. Data from a massive parallel reporter assay served as a training set; this assay allows for the comparison of translation efficiency among tens of thousands of different 5′UTRs in one experiment. It is worth noting that the predictive model in question was tested only for 5′UTR sequences up to 100 bp in length. The obtained predicted MRL values were compared with the results of the experimental assessment of bioluminescence intensity. Spearman’s rank correlation coefficient was calculated for each time point.

### 4.8. Evaluation of the Stability of Secondary Structure of mRNAs

For the mRNAs of various sequences, stability of 3′UTR secondary structure was assessed. The assessment was implemented via the RNAFold method, included in software called the ViennaRNA Package (2.6) [46], and Gibbs free energy (∆G) was computed. Then, the correlation between ∆G and the intensity of luciferase bioluminescence was evaluated.

### 4.9. Cell Type–Specific Expression of RBP Genes

Data on gene expression in cell line THP1 were obtained via processing of raw data from the NCBI Sequence Read Archive (SRR24910839, SRR24910838, SRR24910837, and SRR24910836) [47]. Data on gene expression in DC2.4 cells were obtained via processing of raw data from the NCBI Sequence Read Archive (SRR10967879, SRR10967880, and SRR10967882) [48].

FASTQ files were first subjected to quality control (QC) using FastQC. The FASTQ files were then mapped to reference genome GRCm39 by means of HISAT2 software (2.1.0) [49] with standard settings. Aligned sequence reads were tested for contamination with rRNA and tRNA with the help of the CollectRnaSeqMetrics tool from the Picard Tools software package (3.1.1). Reads were counted by means of the FeatureCounts tool (v2.0.1) [50]. Differential gene expression was analyzed using the R Deseq2 software package (1.18.0) [10.18129/B9.bioc.DESeq2]. Only murine and human homologous genes with 1:1 homology were chosen for the analysis, as determined by Ensembl BioMart Genes version 110 (genes of Homo sapiens, subsection Homologs, Mouse Orthologs). A total of 17,064 homologous genes were selected. For further analysis, we chose genes that are expressed in at least one cell line (transcripts per kilobase million > 1) and have differences in expression at a threshold of *p*_adj_ < 0.05.

### 4.10. Prediction of Binding Sites for RBPs

For predicting RBP-binding sites within the mRNA sequences, the RBPmap web server was employed [51]. The algorithm is based on the weighted-rank approach, which calculates a weighted-rank score for each potential binding site by summing up suboptimal match scores for all motifs within a 50-nucleotide window around a site. The match scores are compared with a background of randomly chosen regulatory regions, and sites with a *p* value < 0.001 are considered significant. Additionally, the algorithm takes into account the clustering propensity of a motif and the overall tendency of regulatory regions to be conserved [52].

### 4.11. Loading of mRNA into LNPs

Encapsulation of mRNA into LNPs was performed as described elsewhere [44]. In brief, a 0.2 mg/mL aqueous solution (10 mM citrate buffer pH 3.0) of mRNA was mixed with an alcoholic solution of a lipid mixture in a microfluidic cartridge on a NanoAssemblr™ Benchtop instrument (Precision Nanosystems, Vancouver, BC, Canada). The components of the lipid mixture were ionizable lipidoid ALC-0315 (BroadPharm; San Diego, CA, USA), distearoylphosphatidylcholine (Avanti Polar Lipids; Alabaster, AL, USA), cholesterol (Sigma-Aldrich; St. Louis, MO, USA), and DMG-PEG-2000 (BroadPharm; San Diego, CA, USA) at a molar ratio (%) of 46.3:9.4:42.7:1.6. The weight proportion of mRNA in LNPs was 0.04%wt. To form particles, we mixed aqueous and alcoholic phases at 3:1 by volume; the overall mixing rate was 10 mL/min.

After that, under sterile conditions, the particles were passed through a filter based on a 0.22 μm PES membrane (Merck, Rahway, NJ, USA) and were stored at 4 °C. Next, the quality of the resulting particles was examined with the help of five parameters: particle size, polydispersity index, zeta potential (Zetasizer Nano ZSP, Malvern Panalitycal, Westborough, MA, USA), mRNA loading, and RNA integrity. 

Particle size proved to be 80 to 90 nm, and the polydispersity index was less than 0.15. Zeta potential was in the range −0.22 to −0.37 mV. Concentration of the mRNA loaded into LNPs was determined via the difference in fluorescent signals after staining with the RiboGreen reagent (Thermo Fischer Scientific; Waltham, MA, USA) before and after disruption of the LNPs. To disrupt the particles, we employed the Triton X-100 detergent (Sigma-Aldrich; St. Louis, MO, USA). The proportion of encapsulated RNA was more than 95%. Analytical characterization of mRNA–LNPs included assessment of RNA quality by capillary electrophoresis using the TapeStation instrument (Agilent Technologies, Santa Clara, CA, USA) after the disruption of LNPs (RNA integrity assessment). Briefly, to lipid particles, we added 2 volumes of 100% isopropanol and vortexed the mixture. Then, it was centrifuged, the supernatant was discarded, and the RNA pellet was washed with 75% EtOH and dissolved in ultrapure water. The area of target peaks of RNA in capillary electropherograms exceeded 85% of the total RNA.

### 4.12. Quantification of a T-Cell Response to the Vaccination

The magnitude of a resultant T-cell response in vaccinated mice was determined by means of the number of splenocytes secreting IFNγ in response to treatment with mycobacterial antigens (a sonicate of *M. tuberculosis*) by the ELISpot method using Mouse IFNg ELISpot Set (BD, Franklin Lakes, NJ, USA) and AEC Substrate Set (BD, Franklin Lakes, NJ, USA) kits in accordance with the manufacturer’s instructions. The sonicate of *M. tuberculosis* was a soluble fraction of an ultrasonic disintegrate of *M. tuberculosis* strain H37RV [53].

Splenocytes were isolated from the spleens of mice in vaccinated and control groups under sterile conditions by a procedure described previously [54]. The splenocytes were seeded at 200 thousand cells per well in ELISpot plates containing a PVDF membrane (BD, Franklin Lakes, NJ, USA) with simultaneous stimulation. The latter consisted of the sonicate of *M. tuberculosis* (10 μg/mL). After the seeding of splenocytes and addition of the stimulus, the total volume of each well was adjusted to 200 μL of the RPMI-1640 culture medium (PanEco; Russia) supplemented with 10% of fetal bovine serum (BIOWEST, Nuaillé, France). Next, the splenocytes were cultured for 20 h in a CO_2_ incubator (5% CO_2_, 37 °C). Cells without the stimulation or stimulated with concanavalin A (Sigma-Aldrich, Darmstadt, Germany) at a dose of 5 μg/mL served, respectively, as negative and positive controls in the ELISpot procedure.

Points corresponding to splenocytes secreting IFNγ were counted on an S6 Ultra device (CTL; Floral Park, NY, USA).

### 4.13. Measurement of the Delayed-Type Hypersensitivity Reaction

Three weeks after vaccination, the delayed-type hypersensitivity reaction of mice in each group (8 mice per group) was assessed as swelling of the left hind paw of a mouse in response to injection with 40 μL of PBS containing 50 IU of purified tuberculin (from the Scientific Centre for Expert Evaluation of Medicinal Products) at 48 h after the injection. The data are presented as Δ (swelling of the left paw minus swelling of the right paw).

### 4.14. Development of a Protective Response after Immunization with the mRNA Vaccines

To induce an experimental tuberculosis infection, a model of intravenous infection of animals was employed; a dose of 500,000 CFUs per mouse was injected 3 weeks after the second vaccination. On day 50 after the infection initiation, the number of mycobacterial cells in the lungs of the infected mice was determined. To this end, the lungs were isolated in a sterile fashion and homogenized in 2 mL of saline, and serial 10-fold dilutions of the lung homogenates were prepared and plated on Petri dishes with Middlebrook 7H10 agar at 50 μL per plate. The Petri dishes were next incubated at 37 °C; after 21 days, macrocolonies of *M. tuberculosis* H37Rv on each plate were counted, and their number per lung was calculated.

### 4.15. Statistical Analysis

Statistical data processing was performed on 5′UTR and 3′UTR datasets as independent samples. All obtained data were not normally distributed, and therefore we chose nonparametric analysis. Normalized data were evaluated by the Kruskal–Wallis test with Dunn’s multiple comparison testing. We assessed correlations using Pearson’s correlation coefficient. The statistical calculations were conducted in the STATISTICA 8 software.

## 5. Conclusions

Altogether, our findings suggest that the selection of UTRs for translation efficiency in vitro and in silico can predict vaccine immunogenicity. TPL as a 5′UTR provides high translation efficiency and may be considered for use in mRNA vaccine studies in murine models. 

## Figures and Tables

**Figure 1 ijms-25-00888-f001:**
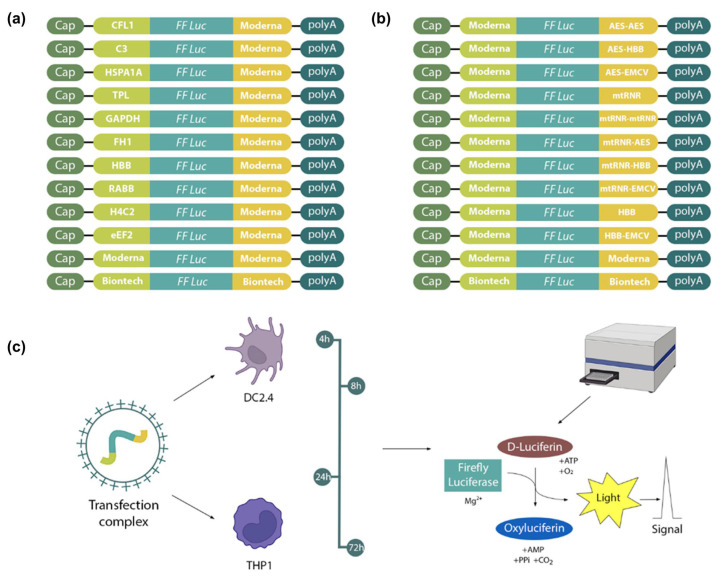
The design of in vitro experiments. mRNA constructs encoding firefly luciferase with various 5′ and 3′UTRs: (**a**) constructs containing various 5′UTRs in combination with the 3′UTR from mRNA-1273 (Moderna); (**b**) constructs containing various 3′UTRs in combination with the 5′UTR from mRNA-1273 (Moderna); and (**c**) bioluminescence of firefly luciferase measured in DC2.4 and THP1 cell lysates at 4, 8, 24, and 72 h after transfection.

**Figure 2 ijms-25-00888-f002:**
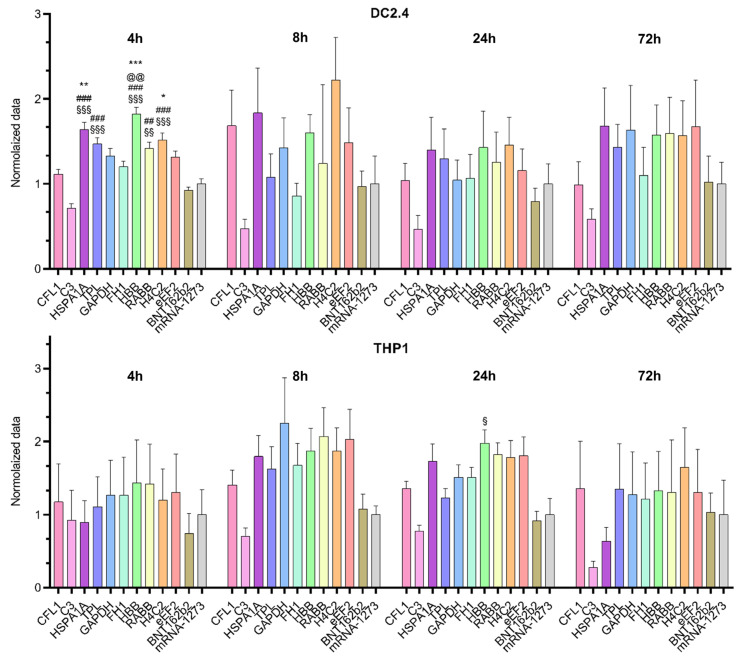
Relative intensity of luciferase bioluminescence in lysates of DC2.4 and THP1 cells transfected with various Luc mRNAs containing different 5′UTR sequences. Data are presented as mean ± SEM. * *p* < 0.05, ** *p* < 0.01, *** *p* < 0.001 as compared with the mRNA-1273 5′UTR; ^##^ *p* < 0.01, ^###^ *p* < 0.001 as compared with the BNT162b2 5′UTR; ^§^ *p* < 0.05, ^§§^ *p* < 0.01, ^§§§^ *p* < 0.001 as compared with the *C3* 5′UTR; and ^@@^ *p* < 0.01 as compared with the *CFL1* 5′UTR. The Kruskal–Wallis test with Dunn’s multiple comparison testing was used.

**Figure 3 ijms-25-00888-f003:**
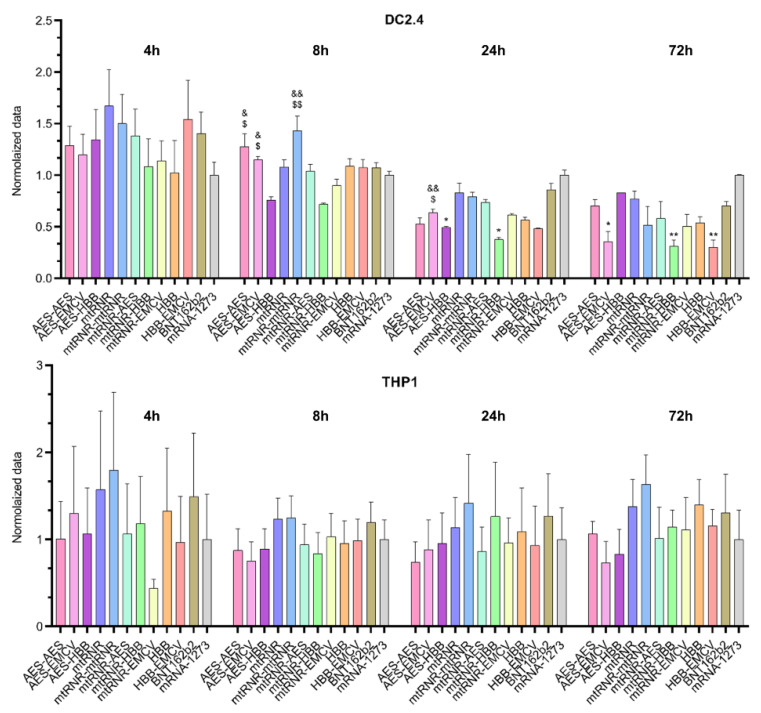
Relative intensity of luciferase bioluminescence in lysates of DC2.4 and THP1 cells transfected with Luc mRNAs containing various 3′UTR sequences. Data are presented as mean ± SEM. * *p* < 0.05 and ** *p* < 0.01 as compared with the mRNA-1273 3′UTR; ^&^ *p* < 0.05 and ^&&^ *p* < 0.01 as compared with mtRNR-HBB; and ^$^
*p* < 0.05 and ^$$^
*p* < 0.01 as compared with AES-HBB.

**Figure 4 ijms-25-00888-f004:**
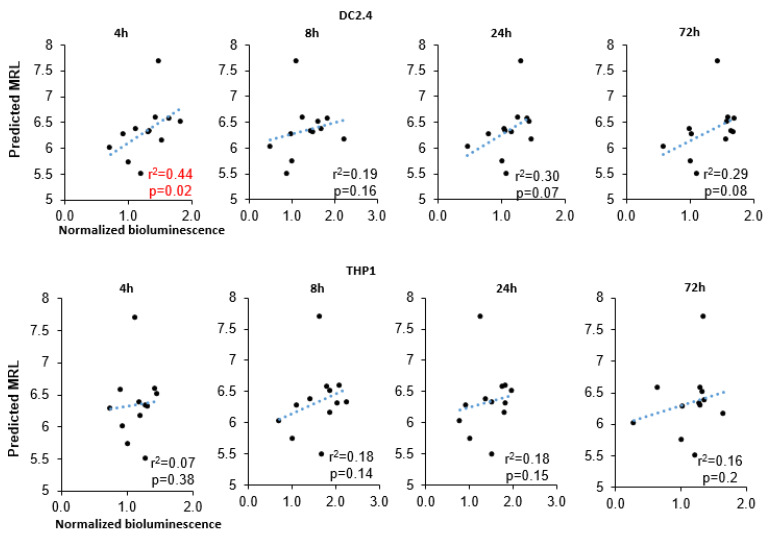
Analysis of correlations between predicted MRL and normalized intensity of bioluminescence.

**Figure 5 ijms-25-00888-f005:**
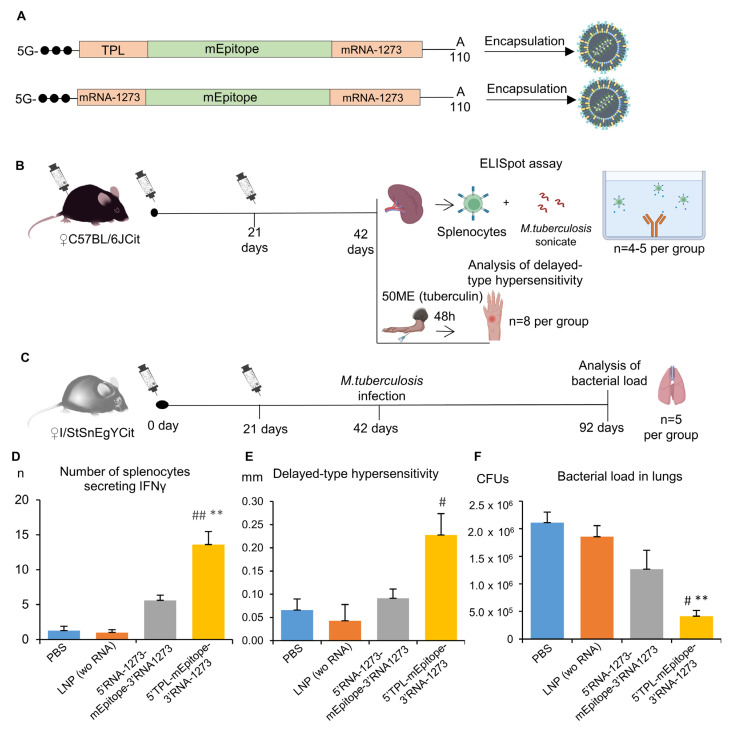
Evaluation of an adaptive and protective immune response after immunization with various mRNA vaccines: (**A**) Schematic representation of the composition of two mRNA-LNP vaccines. One of two different mRNA vaccines (5′-TPL-mEpitope-mRNA1273-3′ or 5′-mRNA1273-mEpitope-mRNA1273-3′), 50 µg of each RNA, was administered intramuscularly. Besides the test groups, there were two control groups in the experiment, the first of which received LNPs without mRNA (in an amount equivalent to ±10% of the number of particles in the RNA vaccine groups), while the second received phosphate buffer; (**B**) Splenocytes from C57BL/6 mice were subjected to the ELISpot assay to count splenocytes secreting IFNγ in response to a sonicate of *M. tuberculosis* and thus to measure the magnitude of the resulting T-cell response; in these mice, we also assessed the delayed-type hypersensitivity reaction after injection of 50 IU of tuberculin; (**C**) In I/StSnEgYCit mice, the influence of mRNA vaccines on bacterial load (*M. tuberculosis*) was assessed in the lungs; (**D**) ELISpot results—the number of cells secreting IFNγ in response to the *M. tuberculosis* sonicate; (**E**) Delayed-type hypersensitivity reaction—differences (∆) between the right and left paw at 48 h after injection of 50 IU of tuberculin; and (**F**) Bacterial load in lungs—the number of *M. tuberculosis* CFUs in lung homogenates 50 days after the infection initiation. The data are presented as mean ± standard error. ** *p* < 0.01 as compared with the PBS group; ^#^ *p* < 0.05, ^##^ *p* < 0.01 as compared with the LNP group.

**Table 1 ijms-25-00888-t001:** Features of the tested 5′UTR and 3′UTR sequences.

5′UTR	MRL	Binding Sites for RBPs (*p* > 0.0001)
CFL1	6.38	**MBNL1**, **SRSF5**, SRSF8
C3	6.02	**PCBP4**, **PUF60**, **RBM6**
HSPA1A	6.58	**RBM22**
TPL	7.70	PABPC3, **RBM22**
GAPDH	6.33	-
FH	5.50	-
HBB	6.52	-
Rabb	6.59	-
H4C2	6.17	-
eEF2	6.31	**EIF4G2**, **FUS**
BNT162b2	6.28	**CNOT4**
mRNA-1273	5.74	**HNRNPH1**, **PABPN1**, **TRA2A**
**3′UTR**	**MFE** **(∆G, kJ)**	
AES-AES	−87	**PCBP2, PCBP4**, **RBM6**
AES-EMCV	−116.7	**PCBP2, PCBP4**, **RBM6**, PABPC3
AES-HBB	−68.7	**PCBP2, PCBP4, RBM6**
mtRNR	−36.9	**QKI, RBM42, RBM45, SF1**
mtRNR-mtRNR	−78.6	**QKI, RBM42, RBM45, SF1,** HNRNPK, **HNRPLL**
mtRNR-AES	−87	**PCBP2, PCBP4, RBM6**
mtRNR-HBB	−67.7	**QKI, RBM42, RBM45, SF1**
mtRNR-EMCV	−111.7	**QKI, RBM42, RBM45, SF1**
HBB	−26.3	-
HBB-EMCV	−102.4	PABPC3, **SNRNP70**
BNT162b2	−86.4	**PCBP2, PCBP4, RBM6**, **QKI**, **RBM42, RBM45, SF1**
mRNA-1273	−31.3	HNRNPK, **PCBP1**, **PCBP2, PCBP4,** RBM23, **RBM6**, **SRSF5**

Proteins whose genes are differentially expressed in DC2.4 and THP-1 cells are highlighted in bold.

**Table 2 ijms-25-00888-t002:** The list of RBP genes differentially expressed between cell lines and effects on the stability of RNAs.

RBP	Role in RNA Stability	Reference
MBNL1	facilitates mRNA decay	[21]
PCBP4	negatively regulates stability of p21 mRNA	[22]
	PUF60 stabilizes Drp1 mRNA	[23]
RBM6	RBM6 as part of a complex with MLKL regulates the mRNA stability of adhesion molecules	[24]
PCBP2	inhibits translation of eIF4G2 and CCNI mRNAs;	[25]
	reduces stability of FHL3 mRNA;	[26]
	necessary for p73 mRNA stability (acts via the CU-rich elements in p73 3′-UTR)	[27]
FUS	stabilizes GluA1 mRNA;	[28]
	stabilizes SynGAP mRNA	[29]
HNRNPH1	stabilizes App mRNA	[30]
QKI	stabilizes miR-20a	[31]
CNOT4	the CNOT4 subunit of the CCR4-NOT complex is involved in mRNA degradation	[32]

## Data Availability

Raw data from this study are available upon reasonable request.

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
