# Peer review of "Untranslated Region Sequences and the Efficacy of mRNA Vaccines against Tuberculosis"

_ijms, 2024, doi:10.3390/ijms25020888_

Round 1
Reviewer 1 Report
Comments and Suggestions for Authors
Dear Research Team,
I am writing to provide feedback on your recent paper that explores mRNA vaccine development targeting Mycobacterium tuberculosis, the causative agent of tuberculosis (TB). Your study focuses on optimizing the untranslated regions (UTRs) of mRNA constructs, a crucial aspect for the stability and translation efficiency of mRNA vaccines. Such optimizations have a direct impact on protein expression levels and, consequently, the immunogenicity of the vaccine.
The development of effective vaccines against TB is of paramount importance, particularly in light of the variable efficacy of the existing BCG vaccine. Your research, with its detailed examination of various UTR sequences, makes a significant contribution to understanding how these sequences influence mRNA stability and translational efficacy. The methodological choice of using luciferase as a reporter gene provides a quantifiable means to assess translational output, which is commendable.
However, the study raises several critical questions and concerns that are vital for refining the research and enhancing its overall impact:
1. UTR and ESAT-6 Antigen Combination: Your effective use of luciferase to evaluate various UTR sequences brings forth a crucial question about the applicability of these findings to the specific context of TB, particularly concerning the ESAT-6 antigen. It is imperative to consider whether the optimal UTR combination identified for luciferase directly translates to the ESAT-6 antigen, a key component of the TB vaccine's efficacy. Proteins can have varying translational requirements, and the response to the same UTR sequences may differ. Direct experimental validation using the ESAT-6 antigen could establish whether the identified UTR combinations are equally effective for TB vaccine development.
2. Codon Optimization in Different Cell Lines: The use of both human-derived THP-1 and mouse-derived DC2.4 cell lines highlights the issue of codon optimization, which can vary significantly between species. This variation may affect mRNA translation efficiency and protein expression. Addressing codon optimization can provide a deeper understanding of the mRNA constructs' behavior across different biological contexts, thereby improving the research's applicability and translational potential.
3. Specificity of Immune Response Assessment: The decision to use whole cell lysate of Mycobacterium tuberculosis rather than purified ESAT-6 protein raises concerns about the specificity and accuracy of the immune response assessment. The complex nature of the whole cell lysate could potentially obscure the specific attribution of immune responses to the ESAT-6 antigen. Employing purified ESAT-6 or other well-defined TB antigens could lead to a clearer understanding of the vaccine's efficacy and immunogenicity, essential for accurately evaluating the vaccine's potential.
4. Temperature Stability of mRNA Vaccines: An essential aspect of mRNA vaccine development, especially for TB, is the formulation's temperature stability. This is critical in regions with limited cold chain infrastructure. While your exploration into mRNA stability provides a foundation, further investigation into how UTR optimizations impact the vaccine's temperature stability is needed. Understanding the extent to which these optimizations can enhance the vaccine's resilience to temperature fluctuations is crucial for its practical utility and accessibility in high TB burden regions.
In conclusion, your research presents promising advancements in mRNA vaccine technology with potential applications for TB vaccination. The suggestions outlined above could lead to the development of more robust, effective, and accessible TB vaccines, thereby making a significant contribution to global health efforts in combating tuberculosis.
Author Response
Reviewer 1
Dear Research Team,
I am writing to provide feedback on your recent paper that explores mRNA vaccine development targeting Mycobacterium tuberculosis, the causative agent of tuberculosis (TB). Your study focuses on optimizing the untranslated regions (UTRs) of mRNA constructs, a crucial aspect for the stability and translation efficiency of mRNA vaccines. Such optimizations have a direct impact on protein expression levels and, consequently, the immunogenicity of the vaccine.
The development of effective vaccines against TB is of paramount importance, particularly in light of the variable efficacy of the existing BCG vaccine. Your research, with its detailed examination of various UTR sequences, makes a significant contribution to understanding how these sequences influence mRNA stability and translational efficacy. The methodological choice of using luciferase as a reporter gene provides a quantifiable means to assess translational output, which is commendable.
However, the study raises several critical questions and concerns that are vital for refining the research and enhancing its overall impact:
Reply: Thank you for your time and effort in reviewing this manuscript. Below, you will find our detailed responses to your comments, along with the corresponding revisions and corrections clearly highlighted or tracked in the resubmitted files. We greatly appreciate your thorough evaluation and valuable feedback.
- UTR and ESAT-6 Antigen Combination: Your effective use of luciferase to evaluate various UTR sequences brings forth a crucial question about the applicability of these findings to the specific context of TB, particularly concerning the ESAT-6 antigen. It is imperative to consider whether the optimal UTR combination identified for luciferase directly translates to the ESAT-6 antigen, a key component of the TB vaccine's efficacy. Proteins can have varying translational requirements, and the response to the same UTR sequences may differ. Direct experimental validation using the ESAT-6 antigen could establish whether the identified UTR combinations are equally effective for TB vaccine development.
Reply: Thank you for pointing this out. As we have used a multi-epitope mRNA vaccine based on ESAT6, commercially available antibodies against ESAT6 are not suitable for assessing the expression level of the product of the mRNA vaccines using different UTRs. Confirming the universality of using UTRs for different coding sequences (CDSs) requires either in vivo immunogenicity studies or the use of different reporters such as firefly luciferase, Renilla luciferase, NanoLuc luciferase, GFP, RFP, etc. It is a distinct and labour-intensive task, which is beyond the scope of this study. However, we agree to mention this as the limitation of the current study in the Discussion section (lines 334-337).
- Codon Optimization in Different Cell Lines: The use of both human-derived THP-1 and mouse-derived DC2.4 cell lines highlights the issue of codon optimization, which can vary significantly between species. This variation may affect mRNA translation efficiency and protein expression. Addressing codon optimization can provide a deeper understanding of the mRNA constructs' behavior across different biological contexts, thereby improving the research's applicability and translational potential.
Reply: We are in complete agreement with your comment. The efficacy of translation can vary both at the cellular level and between organisms. Sometimes results obtained in rodents are poorly replicated in non-human primates and in human clinical trials. Developing a universal optimization algorithm that is effective for the majority of mammalian species and takes into account the nuances of the translational machinery would help to address these challenges in the future.
- Specificity of Immune Response Assessment: The decision to use whole cell lysate of Mycobacterium tuberculosis rather than purified ESAT-6 protein raises concerns about the specificity and accuracy of the immune response assessment. The complex nature of the whole cell lysate could potentially obscure the specific attribution of immune responses to the ESAT-6 antigen. Employing purified ESAT-6 or other well-defined TB antigens could lead to a clearer understanding of the vaccine's efficacy and immunogenicity, essential for accurately evaluating the vaccine's potential.
Reply: Thank you for the remark. The complex composition of mycobacterial lysates, together with the drawbacks mentioned by the esteemed reviewer, offers important benefits when used to test the immunogenicity of experimental vaccines. The majority of tested vaccine candidate antigens are heterologously expressed in various strains of Escherichia coli (E. coli), in spite of the fact that the protein folding and post-translational modifications might be very different compared to the host mycobacterium. Posttranslational modifications such as methylation have been shown to alter the immunogenicity of recombinant proteins when comparisons were made between native mycobacterial protein versus the recombinant protein expressed from E. coli (see for example Temmerman S., Pethe K., Parra M., Alonso S., Rouanet C., Pickett T., Drowart A., Debrie A. S., Delogu G., Menozzi F. D., Sergheraert C., Brennan M. J., Mascart F. and Locht C., 2004. Methylation-dependent T cell immunity to Mycobacterium tuberculosis heparin-binding hemagglutinin. Nat Med 10, 935-41). The purification of native proteins from M. tuberculosis for biochemical and immunological analysis is a complex and laborious process, often resulting in poor yields of protein. Other recombinant expression systems were developed to be used in fast-growing mycobacteria. These systems produce proteins with a glycosylation profile that is closer to, but not identical to, that of the native protein. Given the ultimate goal of developing an anti-TB vaccine, it is our opinion that it is more appropriate to test vaccine-induced immune responses on native antigen preparations.
- Temperature Stability of mRNA Vaccines: An essential aspect of mRNA vaccine development, especially for TB, is the formulation's temperature stability. This is critical in regions with limited cold chain infrastructure. While your exploration into mRNA stability provides a foundation, further investigation into how UTR optimizations impact the vaccine's temperature stability is needed. Understanding the extent to which these optimizations can enhance the vaccine's resilience to temperature fluctuations is crucial for its practical utility and accessibility in high TB burden regions.
Reply: Our stability results indicate that the mRNA vaccine remains stable for at least 6 months when stored at +2 to +8°C. While the 5'UTR has a significant influence on the secondary structure of the RNA molecule, the primary factor affecting the stability of mRNA vaccines is the composition of the storage buffer. However, this is an important point – we will mention it in the discussion and take it into account while planning further studies.
In conclusion, your research presents promising advancements in mRNA vaccine technology with potential applications for TB vaccination. The suggestions outlined above could lead to the development of more robust, effective, and accessible TB vaccines, thereby making a significant contribution to global health efforts in combating tuberculosis.
Reply: We appreciate your evaluation, thank you.
Reviewer 2 Report
Comments and Suggestions for Authors
In this manuscript titled “Untranslated-Region Sequences and efficacy of mRNA vaccine against tuberculosis”, the authors selected various 5’ UTRs and 3’UTRs from eukaryotic genes and tested their abilities upon gene/vaccine translation efficiency in different cell lines comparing with the UTRs achieved from COVID mRNA vaccines. Bioinformatic analysis was carried out to elucidate the reason why these UTRs might affect the translation efficiency in specific cell types. From the library, specific 5’ UTR was further selected to engineer a mRNA vaccine and test its immune response against Mycobacterium tuberculosis in mice. The conclusion in this manuscript can be drawn that UTRs can significantly affect mRNA translation efficiency ex vivo/in vivo, and this effect can be distinct among various cell types. The major concept of this manuscript is clearly delivered, and the article is written with good logic.
It is not a new but crucial topic of study the effect of UTRs over gene expression. One advantage of mRNA-based therapeutics is that there is no necessity to consider fine-tuning the transcription process in the specific host cells while engineering the DNA plasmid affected by these biological components. Direct application of naturally occurring UTRs with good performance for mRNA engineering can circumvent the need to rational design artificial UTRs with additional trial-and-errors. Below are my concerns to the manuscript.
Major concerns:
1. On the result section 3.1 and 3.2, The Figure 3 shown that selected 5’ UTRs have different abilities of impact on luciferase translation in either DC2.4 or THP1 cell lines. However, it is a little hard to read and interpret the results in these figures corresponding to the contents in the manuscript. I believe more information can be extracted from the data with better organization.
i). Following the manuscript description, it may be better to re-organize the UTR luminescence data comparison based on each time point. For example, reconstruction of all the 4-hour data into one figure without the results from the other time points. To my understanding, most of the significant data are from the four-hour time points. This would clearly show and compare among the selected UTRs against the reference control constructs. Same method can be applied to the other data points. The figure content that is not mentioned in the manuscript can be moved to the Supplementary Materials. Also based on the data, is it possible to find out how long can each UTR affect the translation process? It would be interesting to know the duration of engineered mRNA influenced by these UTRs. For example, TPL has the best performance at four-hour time point, while H4C2 shows the highest bioluminescent at 8 hours. Does it mean that 5’ UTRs with longer performance can stabilize the mRNA molecules in specific cell line? The same point can also apply to Figure 4.
ii). It is not clear to me why comparing the data with the 5’ UTR from complement gene C3 in the manuscript (e.g., line 340 and line 344)? Does this UTR have specific functions over translation process? The reference control should be the BNT162B2 and mRNA-1273, correct? I probably would claim that 5’ UTR from complement gene C3 shown negative influence on bioluminescence level compared to the controls at each time point, and in all we can achieve a wide dynamic range with X-fold translation efficiency difference by applying the least to the most functional 5’ UTRs.
iii). Based on the data, HBB shown significant bioluminescence enhancement in both cell lines. Why use TPL instead of HBB for the final animal assay?
iV). In section 3.2, data analysis missing for THP1 cell lines (figure 4, bottom). Some conclusions can be drawn despite all the data shown insignificance. Please add.
2. In my opinion, it is great that this manuscript has experimental data with statistical/bioinformatic analysis (section 3.3 and 3.4) to strengthen their conclusion. I just have some thoughts on these sections.
i) Following the analysis, is it possible to come up with a metric or method that can be used to guide or even predict the functionality of the UTRs in different cell lines? It would be useful if the other researchers can directly apply this tool for rational design their mRNA transcripts.
ii). Do the authors ever consider analyzing (full/partial) 5’ UTR structures’ fold energy (similar to their analysis on 3’ UTR) and their potential correlations to the bioluminescent data? 5’UTR structure can affect ribosome loading during translation initiation and therefore impact the translation efficiency (multiple publications can be found online, e.g., Leppek et al., 2016; Hinnebusch et al., 2016; Dvir et al., 2013).
3. The authors engineered the mRNA vaccine using their best-performed TPL 5’ UTR against tuberculosis and further tested it in mice. As the authors used unmodified uridine in the whole design (line 98-99), I am curious does the vaccine cause strong innate immune response during the assay? If so, how much would this elude the results?
Minor concerns:
1. Line 78, an active “search” should be “research”.
2. Line 119 and line 308 (and maybe other lines), should it be 3 weeks instead of 4 based on the figure 2B?
3. Line 155, based on the cloning method, does the mRNA have additional 5 nt after the polyA tail using SpeI digestion? Would this affect mRNA function? Type IIS restriction enzyme might be a better fit for the cloning process.
4. What purification method the author used to purify their linearized mRNA candidates in the method?
5. Line 341, should it be “figure 3” not “ figure 2”?
6. In figure 6, it would be better to indicate in the x-axis that 5’ UTRs of these sequences (not the vaccine or the gene themselves) are used for this test.
7. The authors used ARCA to synthesize cap-0 mRNA. Do the authors consider the method to synthesize cap-1 mRNA such as CleanCap to further optimize their vaccine translation efficiency? Would this enhance the performance particularly for the cap-dependent translation process? The authors can just address this in the discussion section.
8. It looks like there are additional spaces in line 546 and 548.

My major and minor concerns are in my Review comments. Language wise, please proof read the whole articles. The manuscript is clearly written, but I believe some sentences can be simiplifed.
Author Response
Reviewer 2
In this manuscript titled “Untranslated-Region Sequences and efficacy of mRNA vaccine against tuberculosis”, the authors selected various 5’ UTRs and 3’UTRs from eukaryotic genes and tested their abilities upon gene/vaccine translation efficiency in different cell lines comparing with the UTRs achieved from COVID mRNA vaccines. Bioinformatic analysis was carried out to elucidate the reason why these UTRs might affect the translation efficiency in specific cell types. From the library, specific 5’ UTR was further selected to engineer a mRNA vaccine and test its immune response against Mycobacterium tuberculosis in mice. The conclusion in this manuscript can be drawn that UTRs can significantly affect mRNA translation efficiency ex vivo/in vivo, and this effect can be distinct among various cell types. The major concept of this manuscript is clearly delivered, and the article is written with good logic.
It is not a new but crucial topic of study the effect of UTRs over gene expression. One advantage of mRNA-based therapeutics is that there is no necessity to consider fine-tuning the transcription process in the specific host cells while engineering the DNA plasmid affected by these biological components. Direct application of naturally occurring UTRs with good performance for mRNA engineering can circumvent the need to rational design artificial UTRs with additional trial-and-errors. Below are my concerns to the manuscript.
Reply: Thank you for your time and effort in reviewing this manuscript. Below, you will find our detailed responses to your comments, along with the corresponding revisions and corrections clearly highlighted or tracked in the resubmitted files. We greatly appreciate your thorough evaluation and valuable feedback.
Major concerns:
- On the result section 3.1 and 3.2, The Figure 3 shown that selected 5’ UTRs have different abilities of impact on luciferase translation in either DC2.4 or THP1 cell lines. However, it is a little hard to read and interpret the results in these figures corresponding to the contents in the manuscript. I believe more information can be extracted from the data with better organization.
i). Following the manuscript description, it may be better to re-organize the UTR luminescence data comparison based on each time point. For example, reconstruction of all the 4-hour data into one figure without the results from the other time points. To my understanding, most of the significant data are from the four-hour time points. This would clearly show and compare among the selected UTRs against the reference control constructs. Same method can be applied to the other data points. The figure content that is not mentioned in the manuscript can be moved to the Supplementary Materials. Also based on the data, is it possible to find out how long can each UTR affect the translation process? It would be interesting to know the duration of engineered mRNA influenced by these UTRs. For example, TPL has the best performance at four-hour time point, while H4C2 shows the highest bioluminescent at 8 hours. Does it mean that 5’ UTRs with longer performance can stabilize the mRNA molecules in specific cell line? The same point can also apply to Figure 4.
Reply: Thank you for pointing this out. The figures have been rearranged.
ii). It is not clear to me why comparing the data with the 5’ UTR from complement gene C3 in the manuscript (e.g., line 340 and line 344)? Does this UTR have specific functions over translation process? The reference control should be the BNT162B2 and mRNA-1273, correct? I probably would claim that 5’ UTR from complement gene C3 shown negative influence on bioluminescence level compared to the controls at each time point, and in all we can achieve a wide dynamic range with X-fold translation efficiency difference by applying the least to the most functional 5’ UTRs.
Reply: The objective was to describe all significant changes. You are right; 5’ UTR from complement gene C3 showed a negative influence on the bioluminescence level compared to the controls at each time point.
iii). Based on the data, HBB shown significant bioluminescence enhancement in both cell lines. Why use TPL instead of HBB for the final animal assay?
Reply: We chose TPL as the 5`UTR sequence because it showed the best results in the DC2.4 cell line, had best MRL. We considered data from mouse-derived cell line more relevant for further animal studies. However, we agree that HBB should be highlighted as the promising UTR for human use.
iV). In section 3.2, data analysis missing for THP1 cell lines (figure 4, bottom). Some conclusions can be drawn despite all the data shown insignificance. Please add.
Reply: Much appreciated, the text has been corrected.
- In my opinion, it is great that this manuscript has experimental data with statistical/bioinformatic analysis (section 3.3 and 3.4) to strengthen their conclusion. I just have some thoughts on these sections.
- Following the analysis, is it possible to come up with a metric or method that can be used to guide or even predict the functionality of the UTRs in different cell lines? It would be useful if the other researchers can directly apply this tool for rational design their mRNA transcripts.
Reply: Thank you for bringing this to our attention. Several research groups are currently working on a programme to generate UTRs with specific properties and specificity for a particular cell line. However, these are still in the development stage. Individual studies such as Sample et al, 2019 and Cao et al, 2021 are bringing us closer to this goal.
Sample PJ, Wang B, Reid DW, Presnyak V, McFadyen IJ, et al. (2019) Human 5' UTR design and variant effect prediction from a massively parallel translation assay. Nat Biotechnol 37: 803-809.
Cao J, Novoa EM, Zhang Z, Chen WCW, Liu D, et al. (2021) High-throughput 5' UTR engineering for enhanced protein production in non-viral gene therapies. Nat Commun 12: 4138.
ii). Do the authors ever consider analyzing (full/partial) 5’ UTR structures’ fold energy (similar to their analysis on 3’ UTR) and their potential correlations to the bioluminescent data? 5’UTR structure can affect ribosome loading during translation initiation and therefore impact the translation efficiency (multiple publications can be found online, e.g., Leppek et al., 2016; Hinnebusch et al., 2016; Dvir et al., 2013).
Reply: Yes, you are right. However, we did not conduct a similar analysis. It is believed that 5′UTRs of actively translated mRNAs are shorter on average (Kochetov et al., 1998). Active research into properties of the 5′UTRs affecting translation has revealed some patterns. Strong secondary structure of a 5′UTR with a high GC content is associated with less efficient translation (Leppek et al., 2018; Araujo et al., 2012). Nonetheless, in addition to the negative effects on translation, RNA secondary structures can give rise to higher-order interactions that are conducive to assembly of intermolecular RNA complexes with an RNA-binding protein (RBP) and regulatory RNA molecules.
Therefore, for the comparison of 5' UTR, it is more appropriate to assess MRL.
- Kochetov, A.V.; Ischenko, I.V.; Vorobiev, D.G.; Kel, A.E.; Babenko, V.N.; Kisselev, L.L.; Kolchanov, N.A. Eukaryotic mRNAs encoding abundant and scarce proteins are statistically dissimilar in many structural features. FEBS Lett. 1998, 440, 351–355
- Leppek, K.; Das, R.; Barna, M. Functional 5′ UTR mRNA structures in eukaryotic translation regulation and how to find them. Rev. Mol. Cell Biol. 2018, 19, 158–174.
- Araujo, P.R.; Yoon, K.; Ko, D.; Smith, A.D.; Qiao, M.; Suresh, U.; Burns, S.C.; Penalva, L.O. Before It Gets Started: Regulating Translation at the 5′ UTR. Funct. Genom. 2012, 2012, 475731.
The authors engineered the mRNA vaccine using their best-performed TPL 5’ UTR against tuberculosis and further tested it in mice. As the authors used unmodified uridine in the whole design (line 98-99), I am curious does the vaccine cause strong innate immune response during the assay? If so, how much would this elude the results?
Reply: We did not evaluate the extent to which the innate immune response became activated after vaccination. However, the specific effect of the vaccine may be enhanced by moderate activation of the innate immune response. (Muslimov et al., 2023). Unaltered mRNA vaccines (without pseudouridine) currently in clinical trials, such as those against SARS-CoV2, are still under development. These vaccines are effective at lower doses than modified nucleotide vaccines: CVnCoV [NCT04652102] has a dose of 12μg. Moderna's mRNA-1273 has a dose of 100μg. Since both vaccines tested in vivo had unmodified uridine, we believe that potential activation of innate immunity did not have an impact on conclusions of this study.
Muslimov, A., Tereshchenko, V., Shevyrev, D., Rogova, A., Lepik, K., Reshetnikov, V., & Ivanov, R. (2023). The Dual Role of the Innate Immune System in the Effectiveness of mRNA Therapeutics. International Journal of Molecular Sciences, 24(19), 14820.
Minor concerns:
- Line 78, an active “search” should be “research”.
- Line 119 and line 308 (and maybe other lines), should it be 3 weeks instead of 4 based on the figure 2B?
Reply 1-2: Thank you, the text has been corrected.
Line 155, based on the cloning method, does the mRNA have additional 5 nt after the polyA tail using SpeI digestion? Would this affect mRNA function? Type IIS restriction enzyme might be a better fit for the cloning process.
Reply: After SpeI digestion, indeed, there are 4 nucleotides remaining that are transcribed into mRNA. We believe that this fragment does not affect mRNA function.
What purification method the author used to purify their linearized mRNA candidates in the method?
Reply: The desired length and homogeneity of the synthesized RNA molecules were verified by capillary electrophoresis on a TapeStation. No specific methods were used for RNA purification.
Line 341, should it be “figure 3” not “ figure 2”?
Reply: This was a typo, sorry. Corrected.
In figure 6, it would be better to indicate in the x-axis that 5’ UTRs of these sequences (not the vaccine or the gene themselves) are used for this test.
Reply: Good catch, the figure has been revised.
- The authors used ARCA to synthesize cap-0 mRNA. Do the authors consider the method to synthesize cap-1 mRNA such as CleanCap to further optimize their vaccine translation efficiency? Would this enhance the performance particularly for the cap-dependent translation process? The authors can just address this in the discussion section.
Reply: Thank you, the text has been updated.
- It looks like there are additional spaces in line 546 and 548.
Reply: Thank you, the text has been corrected.
My major and minor concerns are in my Review comments. Language wise, please proof read the whole articles. The manuscript is clearly written, but I believe some sentences can be simiplifed.